# Scanning the Horizon for Environmental Applications of Genetically Modified Viruses Reveals Challenges for Their Environmental Risk Assessment

**DOI:** 10.3390/ijms25031507

**Published:** 2024-01-25

**Authors:** Michael F. Eckerstorfer, Marion Dolezel, Marianne Miklau, Anita Greiter, Andreas Heissenberger, Margret Engelhard

**Affiliations:** 1Umweltbundesamt–Environment Agency Austria (EAA), Landuse and Biosafety Unit, Spittelauer Lände 5, 1090 Vienna, Austria; marion.dolezel@umweltbundesamt.at (M.D.); marianne.miklau@umweltbundesamt.at (M.M.); anita.greiter@umweltbundesamt.at (A.G.); andreas.heissenberger@umweltbundesamt.at (A.H.); 2Federal Agency for Nature Conservation, Division Assessment Synthetic Biology, Enforcement Genetic Engineering Act, Konstantinstr. 110, 53179 Bonn, Germany; margret.engelhard@bfn.de

**Keywords:** genetically modified, virus, bacteriophage, environmental risk assessment, horizon scan, agriculture, nature conservation, veterinary vaccines

## Abstract

The release of novel genetically modified (GM) virus applications into the environment for agricultural, veterinary, and nature-conservation purposes poses a number of significant challenges for risk assessors and regulatory authorities. Continuous efforts to scan the horizon for emerging applications are needed to gain an overview of new GM virus applications. In addition, appropriate approaches for risk assessment and management have to be developed. These approaches need to address pertinent challenges, in particular with regard to the environmental release of GM virus applications with a high probability for transmission and spreading, including transboundary movements and a high potential to result in adverse environmental effects. However, the current preparedness at the EU and international level to assess such GM virus application is limited. This study addresses some of the challenges associated with the current situation, firstly, by conducting a horizon scan to identify emerging GM virus applications with relevance for the environment. Secondly, outstanding issues regarding the environmental risk assessment (ERA) of GM virus applications are identified based on an evaluation of case study examples. Specifically, the limited scientific information available for the ERA of some applications and the lack of detailed and appropriate guidance for ERA are discussed. Furthermore, considerations are provided for future work that is needed to establish adequate risk assessment and management approaches.

## 1. Introduction

Viruses have traditionally played a significant role in biotechnology from its early days to the recent developments of new applications of genetically modified (GM) viruses. Thus, the spectrum of possible use scenarios for GM viruses is very broad. Initially, viruses served as a source of genetic elements, such as the 35S promoter element from the cauliflower mosaic virus, which is used for the construction of recombinant expression cassettes for the development of GM organisms (GMOs) [1]. More recently, highly complex applications based on GM viruses are being developed in the field of synthetic biology (SynBio) [2].

Plant and animal viruses have been used for quite a long time as tools to introduce recombinant DNA constructs into target cells [3] to either create GM cells or to transiently express desired gene constructs in target cells. Examples include the production of transgenic proteins (VAGE—virus-aided gene expression), the silencing of the expression of endogenous genes (VIGS—virus-induced gene silencing), or the modification of the epigenetic regulation of plant gene expression, e.g., via RNA-dependent DNA methylation (RdDM) [4].

More recently, viral vectors have been used to express the molecular components required for genome editing, such as sequence-specific nucleases and guide RNAs, or to introduce repair templates for targeted genome editing, also called virus-induced genome editing (VIGE) [2,5,6]. Such viral vectors can be used to introduce targeted genome modifications in planta, thus avoiding the transformation of isolated single cells or plant tissue in vitro [7,8]. Some VIGE approaches involve no in vitro steps at all, but are based on viral expression constructs that spread systemically in the inoculated plants. Such approaches can also to generate plants with heritable genome modifications [9,10]. Such approaches, which facilitate the genetic modification of whole plants growing in the environment, are significantly different from the traditional methods used to create GM plants. These are based on the regeneration of modified plants from single transformed cells or plant tissue cultures under contained conditions in the laboratory and the selection of particular transformants for subsequent release and propagation in the field. If GM vectors based on viruses, which can be spread horizontally among plants, e.g., by animal vectors like herbivorous or parasitic insects, are used for these applications, whole plant populations could be modified in the environment. Applications of this kind are referred to as “horizontal environmental genetic alteration agents” (HEGAAs) [11]. A proposal by the US Defense Advanced Research Projects Agency to develop HEGAA applications has sparked a debate whether such approaches, which may be associated with a substantial potential for unintended environmental effects and/or misuse, should be pursued [12].

Other environmental applications of GM plant viruses concern the use of plant virus constructs developed to express transgenes in infected plants to protect them against certain plant pathogens, such as bacterial pathogens that are causing devastating diseases in orange groves in the USA [13] or in some European olive-producing areas [14].

GM viruses may also be used for animal biocontrol purposes, building on existing experience with the use of non-modified entomopathogenic viruses for the biological control of agricultural insect pests [15] or insect pests, which are vectors for human pathogens [16]. GM viruses have also been explored for the population control of certain animals, such as foxes, rabbits, stoats, mice, etc., by means of immunocontraception, i.e., applications to induce infertility in pest animals [17,18].

In the context of veterinary medicine, GM viruses are used successfully as vaccine agents or as vector backbones for veterinary GM vaccines [19]. Some of these viral vaccines are designed to be replication-competent and transmissible between animals [20]. Such a transmissible vaccine based on a GM virus was developed to protect European wild rabbit populations against myxomatosis and rabbit hemorrhagic disease. However, the development was discontinued after initial field testing in 2000 [21].

In the EU, GM virus applications are subject to the existing GMO regulations, such as Dir. 2001/18/EC, which requires a mandatory risk assessment to be conducted prior to authorization being granted for environmental releases for either field or clinical testing or for placing GM viruses or products developed from them on the market. However, the multitude of possible applications of GM viruses and the prospects for more GM virus applications being used in the environment is likely to present challenges for the competent authorities and institutions involved in the environmental risk assessment (ERA) of such applications at the national and EU level. Both at the EU level [22] and internationally [2], efforts have been made to identify emerging GM virus applications. The study of van der Vlugt [22], which surveyed developments relevant for the EU, focused on the agricultural applications of SynBio microorganisms with relevance for food production and thus did not provide a comprehensive overview of GM virus applications, specifically with regard to environmental applications.

The study at hand was conducted to provide additional information for regulatory agencies concerning the types of GM virus applications that are being developed for use in the environment and for purposes that may interfere with nature conservation efforts. In addition to a horizon scanning exercise, examples of GM virus applications developed for different purposes were analyzed for possible effects on the environment. Furthermore, the adequacy of the existing risk assessment guidelines in the EU to assess the risks associated with these applications was examined. The goal of this study was therefore to increase the regulatory preparedness in the EU and to raise awareness on issues concerning the risk assessment and management of GM virus applications, which require further attention and efforts.

## 2. Horizon Scan of GM Virus Applications for Release into the Environment

### 2.1. Approach to Surveying the Scientific Literature and Results of the Search

A non-exhaustive survey of the scientific and grey literature was conducted to provide an overview of the current developments concerning GM virus applications with relevance for agriculture and nature conservation and their state of development. The focus of the survey was on GM virus applications developed for different purposes of environmental release.

The survey was limited to publications in the peer-reviewed scientific literature (review and research articles published in the English language) over the past decade (2010–2021). Additionally, we screened reports from EU and international institutions dealing with aspects of the ERA of GM viruses. To survey the scientific literature, the search tool provided by Science Direct was used. A set of general keywords were chosen to focus the search on the topical areas addressed in this study and were combined by Boolean operators. Synonyms that were identified in test searches as strong keywords were included in the search to retrieve relevant publications as indicated in parenthesis below:“genetically modified” (“genetically engineered”, “GM”, “GE”, “gene edited”, “recombinant”);“virus” (“viral vector”, “virus induced”, “virus mediated”, “virus based”, “virus aided”);“environment” (“nature conservation”);“agriculture” (“cultivation”, “crop plant”);“wild animal” (”insect”, “agricultural pest”);“safety” (“biosafety”, “risk assessment”).

The search was conducted as depicted in Figure 1, and it identified 1036 publications. Of these, 696 publications were considered not relevant for further analysis based on the screening of their titles and abstracts. The screening for relevance was conducted according to a similar procedure used by the EFSA [23]. The titles and abstracts of the retrieved references were examined separately by two experts with a background in molecular biology, ecology, and environmental risk assessment to determine whether they were relevant for the project. In case of disagreement, a joint decision was made to include or exclude a specific reference according to the focus of the survey.

The identified publications could be grouped into four broad thematic groups of different types of applications as follows:GM viruses as molecular tools, e.g., transduction or expression vectors, used for the development of GMOs in the lab or in the environment;GM viruses for the development of vaccines for livestock or wild animals;GM viruses developed as agents against plant pathogens in agriculture;GM viruses for use as biocontrol agents for insect pests or invasive alien species.

An overview on the results of the literature survey indicating the numbers of publications found for the mentioned major areas of application is presented in Figure 2.

Additionally, many publications were found that addressed the biological and environmental characteristics of different viruses. These studies were retained to consider whether the information covered in these publications was of potential relevance for the risk assessment of GM virus applications together with the retrieved general reviews on different topics concerning viruses and their environmental interactions.

The studies from the four thematic groups were further checked for whether they addressed specific GM virus applications and were characterized concerning their stage of development according to the following categories:Basic research (BR);Early or advanced research and development stages (R&D);Near-market developments (e.g., applications for field testing) or products available on the market.

For the different thematic groups, specific criteria were used to assign a stage of development.

For the first group, i.e., GM viral tools or vectors, near-market/market status was assigned for tools that are already distributed commercially and where the respective tools can be purchased from companies that supply biotechnological tools and reagents. R&D status was assigned for GM vectors or viral tools that were previously developed and reused in the retrieved studies or for vectors that are shared among laboratories and applied by different research groups.

For the second group, i.e., GM viral vaccines or GM viruses used to develop vector vaccines, near-market/market status was assigned to commercial vaccine products already available or vaccines that have been tested in clinical trials for a subsequent authorization of wider use. R&D status was assigned for vaccine applications that were already tested for their safety and for efficacy to elucidate an immunological or protective effect, i.e., in cases when the reported data obtained from small groups of tested animals suggest that further (clinical) testing may be warranted. Studies that described initial development work towards the creation of candidate vaccines were considered to be basis research.

For the last two groups, i.e., GM viruses used against pathogens and in other biocontrol applications, no commercially available applications were found in the survey; information indicating that field trials were conducted was used as the criterion for near-market developments. The assignment of either BR or R&D status was difficult for these groups. Studies that demonstrated the applicability of a specific GM virus construct to address a relevant agricultural or biocontrol problem were categorized into the R&D category; studies that tested different constructs but did not focus on a specific application for which a proof of principle was obtained were regarded as BR.

In some cases, information obtained from the grey literature for the respective applications was taken into account to determine the stage of development of certain GM virus applications. The grey literature also identified additional GM virus applications that were not retrieved by the literature search. Table 1 summarizes the results of the survey of the peer-reviewed scientific literature only.

As indicated in Figure 2 and Table 1, not all of the publications that were considered relevant actually described GM virus applications. Some studies addressed viruses that were (genetically) modified by methods other than GM technology, including the introduction of targeted mutations by techniques such as genome editing. In virology, genetic changes are commonly introduced into viral strains by means of serial passaging, heat/cold treatments, or other methods of attenuation that promote the creation of progeny with genetic variability. Such attenuated strains are traditionally used for certain virus applications, e.g., for vaccine development. Other papers indicated the use of GM techniques as a possibility of future developments but neither described specific GM virus applications nor referred to publications describing specific GM virus applications.

### 2.2. GM Virus Applications Identified in the Surveyed Literature and in Additional Sources

A general observation could be made for all the groups of applications. GM virus applications are mostly based on classical molecular cloning techniques or a combination of different genetic elements, e.g., by recombination. In contrast to GM plants and animals, genome editing or other new genomic techniques (for a description of such techniques and their possible effects, see [24,25]) are used less frequently for the genetic modification of viruses. As discussed, e.g., by Okoli et al. [26], genome-editing approaches are used for the development of GM vaccines, e.g., to generate recombinant mutants to inhibit viral replication. However, viral vectors are important tools for the creation of genome-edited organisms, in particular genome-edited plants, as explained in the following chapter.

#### 2.2.1. GM Virus Vectors/Tools

The list of publications retrieved for this group is provided in Appendix A. In general, two main groups of applications were distinguished. The first group (19/31 studies) comprised the use of virus vectors to facilitate the (transient) silencing of the expression of endogenous genes, mostly to investigate the biological functions of these genes. A total of 16 of these virus-induced gene silencing (VIGS) studies used vector systems based on modified tobacco rattle virus (TRV) in different plant species, including arabidopsis, tomato, tobacco, pepper, Centaurea spp., Chinese narcissus, Lilium, and Forsythia. Many of these studies utilized the pTRV1 and pTRV2 vectors initially developed by Liu et al. [27] or TRV vectors prepared by Chen et al. [28].

The other main group of applications (12/31) consisted of approaches to using viral vectors for the expression of heterologous proteins in animals or animal cells via virus-aided gene expression (VAGE). Most of the identified applications (8/12) described the design of expression vectors based on modified nucleopolyhedroviruses (baculoviruses). Such vectors are used to facilitate the production of subunit vaccines, protein nanoparticles, and virus-like particles (VLPs), particularly in insects and insect cell cultures [29,30].

In addition to VIGS approaches, applications for the targeted modification of plant genomes were developed. These approaches are referred to as virus-induced genome editing (VIGE) and are increasingly being used for the introduction of targeted mutations into a broad range of different plant species [2,6,31]. Certain viral vectors also allow for the expression of larger-sized recombinant constructs, which facilitates the systemic expression of all components needed for the editing of specific genomic sequences in previously unmodified plants [10]. For approaches that employ viral vectors with a limited cargo capacity but a sufficiently broad host range like TRV, hypercompact CRISPR-Cas nucleases, such as Cas12f1, can be used [32]. In other instances, GM viral vectors developed from TRV [9] or potato virus X (PVX) [33] are used to express (multiple) guide RNAs in GM plants harboring a transgenic Cas9 nuclease. These GM vectors are able to infect both somatic and generative plant tissues and can thus introduce heritable targeted mutations into the genomes of the host plants.

If viral vectors are only used for production purposes in contained-use facilities, no substantial environmental exposure would be expected. However, in the case of such viral vectors being disseminated to whole plants growing in the field, e.g., via spraying or transmission by insect vector species, e.g., as proposed for HEGAAs, significant levels of environmental exposure are to be expected. Such scenarios raise significant concerns regarding their safety and the possibilities of misuse [11,12].

#### 2.2.2. GM Vaccine Agents/Vector Vaccines

All the publications retrieved for this group of applications are listed in Appendix A. Most of these studies addressed the use of different viruses as vaccines for husbandry animals and pets. However, only 60% of the studies (69/116) described GM virus applications. The other applications used natural virus isolates or virus strains that were modified by methods such as attenuation via passaging or other treatments, which favor the formation of genetic variants.

The retrieved studies covered developments of various vaccine types for a wide range of animal species, including poultry (21 applications, including 1 study describing a clinical trial) and fish species (15 studies for different fish such as carp species, including koi, turbot, tilapia, Singapore grouper, flounder, and trout). Other developments included vaccines for pigs (15 studies, including 2 advanced applications of a chimeric GM porcine reproductive and respiratory syndrome virus (PRRSV) vaccine [34] and a GM PRRSV-based vaccine, which expresses a structural transmembrane glycoprotein derived from classical swine fever virus (CSFV), a main immunological target of CSFV [35]) and cattle (12 studies for vaccine developments, mostly for foot and mouth disease virus (FMDV)). Only a few applications were retrieved for dogs, cats, sheep, and rabbits. A single application was also retrieved for a subunit vaccine expressed by GM baculovirus targeting a wild animal, i.e., Steller sea lions [36].

These applications differed substantially whether they resulted in the release of GM viruses or GM materials and in their ability to spread in the environment. Several types of vaccines (inactivated vaccines, live attenuated vaccines, subunit, or VLP vaccines) do not involve the intentional (environmental) release of GM virus constructs. Also, replication-deficient GM viruses and GM vector vaccines are expected to result in limited environmental exposure following vaccination. An example for the latter type of vaccines is a development that was reported in the grey literature: a GM vaccine against devil facial tumor disease (DFTD) in Tasmanian devils (Sarcophilus harrisii) developed by the University of Tasmania (Australia). An application for a test trial was notified to and risk-assessed by the Australian Office of the Gene Technology Regulator in March 2023 [37].

Replication-competent GM vaccines maintain the capacity for further transmission upon release into the environment. Besides other approaches, such GM vaccines are used to prevent or treat zoonotic diseases [38]. Ongoing research in this area is exploring the development of replication-competent virally vectored vaccines with the potential for transmission throughout populations of wild animals [39]. One such transmissible GM vaccine is based on a species-specific cytomegalovirus (CMV) [40] and was developed to reduce the reservoir of viral pathogens for emerging zoonoses (e.g., Lassa fever virus) in animal populations, such as African rodents (mastomys rats), which cannot be easily reached by conventional, non-transmissible agents.

#### 2.2.3. GM Viruses against Plant Pathogens

The literature survey conducted in the framework of this project retrieved only a few publications addressing the use of GM viruses against plant pathogens. For a list of the identified publications, see Appendix A. Two publications described the use of CTV-based vectors for the expression of heterologous genes in plants [41,42]. One of these described a VIGS approach to create a “trap crop” for *Diaphorina citri*, the insect vector transmitting the pathogen, which causes citrus greening disease. The modified citrus trees are more attractive for the vector insects, which can be selectively removed [42]. The other retrieved studies did not involve GM viruses.

However, additional sources indicated further applications based on GM CTV vectors. One of these approaches explored the use of a GM CTV vector to express parts of a *D. citri* gene to induce RNAi-mediated interference with vital functions of the insect vector [43]. Another example of a fairly well-developed application concerned the use of GM citrus tristeza virus to prevent and treat citrus greening disease in orange trees. This application was risk-assessed for field trials in Florida, USA, in 2020 [44].

#### 2.2.4. GM Viruses as Biocontrol Agents

The literature search retrieved 38 publications, which are listed in Appendix A. The identified studies described applications of different viral agents for biocontrol purposes, mostly focused on the control of insect pests in agriculture. Of these, 22 studies used different baculoviruses, mostly nucleopolyhedroviruses associated with different lepiopteran pest species. In addition, a densovirus (JcDV) was investigated to control the lepidopteran pest *Spodoptera frugiperda* [45].

Many of the retrieved studies were exploratory, i.e., in the BR or early R&D stages, and only two studies tested non-GM baculovirus isolates in field trials [46,47]. Most of the studies investigated natural isolates of the listed viruses. Only very few of the studies used GM viruses, mostly in the early research stages (BR). One single R&D-stage study described the development of Chilo iridescent virus (CIV), an iridovirus with a broad host spectrum, into a more potent control agent by the insertion of a toxin gene [48].

Two studies addressed non-GM bacteriophages (Listeriaphage P100 and Bacteriophage LPST144) as examples of control agents for food pathogens in food production. Sagona et al. [49], Gibb et al. [50], and Huss and Raman [51] discussed the use of GM bacteriophages for the following purposes:To alter the host-specificity, host-range, and host-killing ability of phages;As vectors to express antimicrobial substances;In human therapeutic and veterinary applications as well as for industrial applications (e.g., to reduce the occurrence of pathogens in food production);For detection and biocontrol purposes.

A particularly elaborate approach for the latter purpose is a proposal to use lytic GM bacteriophages to protect or cure crop and horticultural plants (e.g., olive trees and vine) against a bacterial plant pathogen, *Xylella fastidiosa*. The approach was designed by a consortium from Wageningen University, NL, and is still in the conceptual stage. However, it would involve the exposure of the environment to different GM microorganisms. A detailed discussion of this applications is presented below in Section 3.3.

A review paper retrieved during the horizon scan [52] described another biocontrol approach that is relevant in terms of the environmental use of GM virus applications. The review covered biological control strategies for immunocontraception, which are based on GM vaccine agents. Such strategies were explored for the management of vertebrate pests, such as rodents, rabbits, cane toads, and carp, in Australia [53]. However, major challenges regarding efficacy (i.e., insufficient duration of sterility) and delivery (i.e., ineffective transmission of the viral vector) were reported for such applications [54,55,56,57]. Most of these approaches have been abandoned in the meantime, and only the use of pathogenic non-GM viruses to control invasive fish species, such as tilapia or carp, is still being evaluated [58,59].

## 3. Considerations for the Assessment of Environmental Effects of GM Virus Applications

Applications of GM viruses are subject to different regulations. For example, in the EU, there is the current legislation for the use of GMOs in contained use (Dir. 2009/41/EC) and for deliberate release and placing on the market (Dir. 2001/18/EC) as well as the regulations for veterinary medical products (Reg. (EU) 2019/6). These regulations mandate requirements for the risk assessment of GM products before their use may be authorized. Dir. 2001/18/EC provides the general framework for the environmental risk assessment (ERA) of GMOs that are notified for environmental release for experimental trials or for placing on the market of GM products. For veterinary medical products that are GMOs or contain GMOs, Reg. (EU) 2019/6 requires an ERA according to the requirements stipulated in Dir. 2001/18/EC. This ERA comprises a comprehensive case-specific assessment of the potential adverse environmental effects that could result from the characteristics of the specific GM viruses, the exposure of the receiving environments, and from the interactions with target or non-target organisms in these exposed environments. The ERA required by the EMA is considered to provide an in-depth and complex assessment [60].

Several examples for applications are discussed in the following chapters to illustrate the specific scenarios of use and exposures and to identify the potential effects that need to be considered during a risk assessment. The examples for such use cases were based on the different types of applications that were identified by the horizon scanning and additional (regulatory) information on newly developed applications of GM viruses. In particular, we selected applications that could potentially affect biodiversity or nature conservation and could result in negative effects on human and animal health. Earlier considerations regarding the ERA of GM virus applications, e.g., GM virus vectors for vaccine development [60,61,62], were taken into account for the analysis of the following examples:A GM virus vector vaccine against communicable facial cancer in Tasmanian devils;A GM citrus tristeza virus expressing defensins to protect orange trees from a bacterial disease (citrus greening disease);GM bacteriophages for the biocontrol of the plant pathogen *Xylella fastidiosa*.

These examples were chosen with a view to cover the recent developments in several fields of application, such as agricultural applications, including the control of pathogens affecting agriculturally relevant plants (e.g., orange trees and olive trees) as well as veterinary applications targeting threatened wild animals of conservation concern, such as Tasmanian devils. Two of the chosen examples are currently undergoing field testing based on authorizations for deliberate release (GM citrus tristeza virus and GM virus vector vaccine). The third example (GM bacteriophages for biocontrol) represents a proposal for a future use scenario that would involve the spread of the released GM phages into the environment and could lead to a wider and less predictable pattern of environmental exposure.

### 3.1. GM Virus Vector Vaccine against Facial Cancer in Tasmanian Devils

#### 3.1.1. Description of the Application

The GM vaccine used in this application (Wild Immunity Vector Adenovirus 20-WIVA20) consists of a replication-defective human adenovirus serotype 5 (HAdV-5) vector, which was modified to express tumor antigens to protect Tasmanian devils against devil facial tumor disease (DFTD).

The GM adenovirus construct was prepared using a commercially available platform that allows customized DNA fragments derived from the HAdV-5 genome to be modified and assembled into a linear double-stranded DNA construct [63]. Adenoviruses can infect a wide variety of vertebrates; human adenoviruses have a host range limited to only mammals [37].

The following modifications were introduced into the GM HAdV-5 vector virus:The HAdV-5 early transcribed E1 region was deleted. The E1 genes are essential for viral gene expression and replication, and the deletion of these genes prevents the multiplication and expression of late-transcribed genes, thus limiting the production of infectious virus particles [64];The early-transcribed E3 region was deleted as well. This region contains genes that are necessary for wild-type adenoviruses to evade host immune responses. The deletion of the E3 region results in an increased immunogenic response in host animals to the GM virus;An expression cassette was inserted into the early transcribed region to express two antigen constructs designed to induce immune responses against DFTD tumor cells. One antigen was designed to elicit a response against DFT1 tumor cells; the second antigen was a construct that contained 18 short epitopes specific to DFT1 and DFT2 tumor cells (i.e., a polypeptide neoantigen) and a fluorescent marker gene.

The GM WIVA20 virus can express the antigen genes in the target animal [65], but it cannot replicate or cause a disease [37]. The expression of the GM antigens is transient, as the GM vaccine would be cleared by the host immune system within days or weeks, except in case of integration of the viral DNA into the host genome, which is considered to be a very rare event [66].

The obtained GM vector virus is similar in structure to other previously designed HAdV-5-based vaccines, for example, vaccines that are used in humans (e.g., the Vaxzevria COVID-19 vaccine produced by AstraZeneca), vaccines used for veterinary purposes (e.g., the ONRAB^®^ rabies vaccine authorized in North America, which is based on a replication-competent GM HAdV-5 vector), and vaccines used in cancer immunotherapy [37].

#### 3.1.2. Exposure Pathways and ERA Considerations

For the inoculation of the test animals, the GM WIVA20 vaccine was produced in the form of infectious viral particles in helper cells and was administered via intramuscular or intratumoral injection or by direct instillation into the oral cavity to test different methods of inoculation [37]. After inoculation, the GM viral particles may be distributed in the body of the vaccinated animals and can be excreted or shed. Such shedding is considered the main route of environmental exposure in the case of replication-incompetent adenovirus-based vaccine agents [37]. The amount of shed virus particles, however, would be limited by the dose of the viral vaccine that was used for the vaccination. Accidental exposure of human staff to the vaccine agent itself or to the vaccinated animals needs to be considered, as does the exposure of non-target animals to vaccinated Tasmanian devils and to virus particles that are shed by them, e.g., via saliva, feces, or, to a lesser degree, urine.

The following risk issues were considered for the risk assessment of the time-limited, partly contained test trial [37]:Infection of human staff who are accidentally exposed to infectious material during immunization or the handling of immunized animals. Due to the characteristics of the vaccine, infections would not be sustained over longer periods, and then only limited health effects, if any, were to be expected [37];Infection of non-target animals after contact with vaccinated Tasmanian devils and with saliva or feces from vaccinated animals that contain virus particles. While contact with other mammals was restricted in the test facilities, contact of birds with vaccinated animals or shed virus particles would be plausible. However, the effects were considered to be minimal, as HAdV-5 is different to adenoviruses occurring in birds and exposure is not believed to lead to adverse effects according to the OGTR [37];Generation of replication-competent GM viruses could happen via recombination or complementation in animals or humans infected with the GM virus and other AdV variants. However, these scenarios were considered highly unlikely by the OGTR due to the properties of the AdV and the multiple modification events necessary. Also, the likelihood that co-infection and complementation events would occur in the cells of the vaccinated Tasmanian devils or other exposed animals or humans would be very small under the conditions of the trial. Similar conclusions were drawn for recombination events that could also lead to different combinations of GM or wild-type viruses after multiple recombination events, including to GM viruses with an altered host range [37].

Concerning unrestricted release scenarios, the following aspects would need to be considered:

The GM virus could be present in inoculated animals (or tumors) for several weeks or even months. There is some uncertainty about the possible duration and extent of shedding and the likelihood of a possible infection of other vertebrate hosts (including people, horses, cattle, pigs, sheep, goats and domestic fowl, wild birds, bats, and reptiles). In addition, AdVs can remain infectious for long periods in the environment, for weeks in tap water, sewage effluent, and sea water [67], and for from 7 days to 3 months on dry surfaces [68].

There is some uncertainty concerning the stability of the GM virus. Some mutations were detected in the GM virus that was produced for use as the vaccine. The number of these mutations were in line with the mutation frequencies that can be expected for the replication of HAdV-5 viruses [69]. According to an analysis by the applicant, the identified mutations did not impact functions of the GM WIVA20 vaccines [37]. Another relevant issue involves possible recombination events upon the co-infection of humans and animals with GM WIVA20 as well as other AdVs. Such viruses, including non-human AdVs, would likely be present in the environment. However, the environmental spread of recombinant viruses expressing GM immunogenic peptides will be limited if the expression of the GM insert has a negative impact on viral replication, i.e., by eliciting an immune response. GM inserts in recombinant vector vaccines that are not beneficial for the propagation of the GM virus will likely get lost during sequential viral replication/hybridization events [70]. The interplay of the factors influencing transgene stability, such as the characteristics of the modified vector genome and the inserted transgenic sequences, the expression level of the transgene(s), the host cell environment, and the virus abundance, is complex, poorly understood, and challenging to predict [61,71]. The ERA requirements, according to Dir. 2001/18/EC, do not mandate the provision of (WGS) data to address this issue appropriately [61].

During the trial, the risks resulting from the co-infection of Tasmanian devils or other host species with multiple adenovirus strains and the recombination of naturally occurring adenoviruses with the GM WIVA20 virus were considered to be negligible by the OGTR due to the conditions imposed by the regulator. These conditions mandate that freshly vaccinated animals are kept in confinement for the time when shedding occurs and are only relocated when tests indicate that shedding has stopped.

There are uncertainties concerning potential effects for non-target organisms infected with GM WIVA20 expressing polypeptide neoantigens. The used promoter could drive the expression of recombinant proteins in a wide range of mammalian cells. A potential cross-reactivity with healthy tissues of humans, birds, or other animals could occur with a low risk of autoimmunity in humans and animal species, as concluded by OGTR [37].

### 3.2. GM Citrus Tristeza Virus to Protect Citrus Trees against a Bacterial Disease

#### 3.2.1. Description of the Application

Citrus greening disease, or Huanglongbing (HLB), is a bacterial disease (caused by *Liberibacter asiaticus*) affecting citrus and orange trees [72]. The disease and its vector species, psyllids (e.g., *Diaphorina citri*), were introduced into the Americas in the 2000s. Since 2005, the disease has caused significant damage to commercial orange and citrus fruit production in the USA, e.g., in Florida [73]. The pathogenic bacterium colonizes the phloem of infected trees and inhibits the flow of essential nutrients in the vascular system. This leads to damage in the root system as well as subsequent damage to leaves and fruits. The pathogen is spread efficiently by its highly mobile and fast-reproducing vector, *Diaphorina citri* [74], which shows increased fitness and fecundity upon infection with *Liberibacter asiaticus* [75].

To protect orange trees, an approach was developed based on a GM plant virus vector derived from modified CTV isolates to express a variety of antimicrobial proteins (defensins) in citrus trees. CTV is an RNA plant virus (from the genus *closterovirus*) that can infect various citrus plants. The virus is associated with the phloem of the host plant and multiplies in the cytoplasm of phloem parenchyma cells [76]. The GM CTV vector is derived from wild-type CTV strains (T30 and T36), which occur naturally in Florida and do not lead to severe disease symptoms (stunting, slow or quick decline, stem pitting) in the orange varieties grown in Florida. The genes expressing the defensins (SoD2, SoD7, SoD8, SoD9, SoD11, SoD12, and/or SoD13) were derived from spinach. Previous studies, e.g., to create HLB-resistant GM orange trees, indicated that these defensins can protect orange trees against infection with *L. asiaticus* and thus against HLB disease [44,77].

#### 3.2.2. Exposure Pathways and ERA Considerations

For the treatment, scions (i.e., stem, leaf, or bark pieces) infected in the laboratory with CTV-SoD virus were grafted onto healthy orange trees in orange groves and tree nurseries or orange trees that were already infected with Liberibacter asiaticus. Upon inoculation, the GM CTV-SoD virus spread systemically in the vascular tissue of the infected citrus trees and expressed the defensins in the very plant tissues that were colonized and affected by *Liberibacter asiaticus*. CTV-SoD is expected to persist for longer periods (of up to several years) in infected citrus plants. It is not expected to introduce genetic changes into the genome of treated orange plants.

No other forms of release of the GM CTV-SoD virus, e.g., spreading by infected vector insects, are foreseen, and such exposure pathways were consequently not addressed by the USDA in its assessment [44]. However, the possible spread of CTV-SoD by vector insects, starting from inoculated trees, would lead to a similar result and needs to be taken into account when analyzing additional exposure pathways and impact areas.

In case the exposed areas are indeed limited to orange trees deliberately inoculated by humans, the range of exposed organisms would include the target pathogen and non-target organisms that come into contact with GM CTV-SoD-infected orange trees, including non-target bacteria and fungi as well as non-target animals such as psyllids, aphids, mites, and nematodes. If the further spread of CTV-SoD by (insect) vectors, either accidentally or via the unauthorized movement of infected scions, were to occur, the potentially exposed areas would comprise further areas and possibly also other (citrus) host plants and a wider range of associated non-target species.

The relevant issues to be considered for the risk assessment of GM CTV-SoD are as follows:Adverse effects may occur through changes in the infectivity and pathogenicity of the GM CTV virus as well as through changes in its host range. Evolutionary changes due to the higher mutation rates in RNA plant viruses like GM CTV-SoD as well as through recombination/complementation can be expected. Indications of such changes can be provided by a thorough molecular characterization of the GM virus [78]. Based on a comparison of CTV-SoD with the CTV strains endemic in Florida (in particular, the CTV strains T30 and T36), the USDA concluded that CTV-SoD does not show novel biological characteristics [44]. However, for an application of GM CTV-SoD in the EU, the CTV strains that occur in Europe would need to be considered [76];Additional environmental hazards could arise due to the spreading and establishment of the GM virus in other host plants outside of the areas of intended use. For this scenario, GM CTV-SoD would have to spread from the inoculated trees into the surrounding ecosystems via parts of the GM-virus-infected plants or by transmission through animal vectors. Uncertainties exist whether this is possible. In Florida, transmission by the CTV vector insects was not detected [44]. However, the complementation of poorly transmissible CTV strains (T36) with other naturally occurring CTV strains in the laboratory did increase the ability of CTV-SoD to be transmitted by aphids, such as the main US vector of CTV, the brown citrus aphid (*Toxoptera citricida*) [79]. In case the insects known to vector CTV in Europe [76] could also transmit GM CTV-SoD, the possible spread to other citrus plants relevant in the Mediterranean region of Europe would need to be investigated. This would concern plants such as lemon, lime, sweet and sour orange, mandarin, tangerine, grapefruit, trifoliate orange (*Poncirus*), and dwarf orange (kumquats);It is also uncertain whether pre-infection with naturally occurring CTV variants could protect possible EU host plants from additional infection by GM CTV-SoD due to the current limited understanding of cross protection (superinfection exclusion) [80];Regarding the effects on the target organisms, it needs to be assessed whether the target organism *Liberibacter asicaticus* can develop resistance to the spinach defensin(s). Genetic instability of the transgene(s) in GM CTV-SoD could also lead to a loss of efficacy to control the target pathogen and has to be considered, as does the evaluation of the efficacy of the GM CTV-SoD application;Possible effects on non-target organisms were not assessed comprehensively by the USDA [44]. Their assessment focused on the experience associated with the human consumption of spinach, which indicates no antinutritive effects of the defensins as part of a diet including spinach [81]. However, spinach defensins can have adverse effects on exposed non-target bacteria. The defensins were shown to exert an antimicrobial effect on Gram-negative bacterial species and a less strong effect on fungi (e.g., *Fusarium*) and Gram-positive bacteria [82]. However, the mode of action of their effects is currently not well understood;Other non-target organisms could also be exposed to the spinach defensins upon feeding on orange trees infected with GM CTV-SoD. A report by the National Academies of Science [83] indicated a wide variety of species that could be exposed and possibly affected in the USA, including the citrus leaf miner (Phyllocnistis citrella), several species of scale insects (Unaspis citri, Chrysomphalus aonidium, Lepidosaphes beckii) and whiteflies (Dialeurodes citri, D. citrifollii, Aleurothrixus floccosus, A. woglumi), the citrus mealybug (Planococcus citri), root-associated beetles (Diaprepes abbreviatus, Pachnaeus litus, Pachnaeus opalus), citrus mites (Aculops pelekassi, Phyllocoptruta oleivora, Eutetranychus banksi), and nematodes (Tylenchulus semipenetrans, Radopholus similis, Belonolaimus longicaudatus, Pratylenchus coffeae, P. brachyurus). Related species should be considered as non-target organisms in Europe.

The assessment of the possible environmental effects of the GM CTV-SoD virus is associated with a variety of relevant uncertainties. It is currently unclear whether the risk assessment conclusions relevant for the USA [44] are transferrable to other regions such as Europe, e.g., also considering uncertainties regarding target and non-target organisms as well as the range of vector insects occurring in the EU. The results of the ongoing field trials and the studies supporting the risk assessment by the USDA need to be analyzed regarding whether they provide relevant data for a number of risk hypotheses, such as possible adverse effects of the transgenic defensins on non-target organisms or the possible spread to other CTV host plants. The EFSA noted that the US assessment [44] did not cover all the risk areas that would need to be considered in an environmental risk assessment in the EU under Directive 2001/18/EC [78].

A possible transmission of the GM virus by vectors to other host plants would significantly increase the area impacted by the GM virus and the citrus species, which would need to be monitored. Furthermore, the spread of the GM virus outside areas of commercial cultivation of orange trees would result in the exposure of natural habitats and ecosystems. The exposure of relevant non-target organisms in EU environments has not been conclusively clarified yet and requires further assessment.

### 3.3. GM Bacteriophages for the Biocontrol of a Plant Pathogen

#### 3.3.1. Description of the Application

The proposal for the “Xylencer” application [84] represents a particularly elaborated approach to using GM bacteriophages to protect or cure crop and horticultural plants, e.g., olive trees, from a bacterial plant pathogen, *Xylella fastidiosa*. The development the application is in the early research stage; several aspects of its application have been developed in laboratory studies.

The approach consists of two different interacting elements:A set of lytic bacteriophages, which were shown to target the bacterial pathogen *X. fastidiosa* [85,86]. These phages were genetically modified to express a peptide derived from the flagellin protein (flg22) of *X. fastidiosa*, which triggers defense response mechanisms in infected plants. The GM phages also express a chitin-binding protein (PD1764sh). This protein was fused to the capsid protein of the GM phages and mediates adherence to sap-feeding insects for dissemination;A bacterium developed from a strain of *Xanthomonas arboricola* was genetically modified and acts as a phage-delivery bacterium (GM PDB). *X. arboricola* is a recognized phytopathogen (EFSA PLH 2014) and can cause canker and bacterial blight, e.g., in stone fruits, almonds, and walnuts [87]. For the Xylencer approach, a non-pathogenic strain, *X. arboricola* CITA 44, was used to construct the GM PDB. The GM bacterium was modified with a sensing mechanism in order to express the GM bacteriophages selectively in crop plants infected by *X. fastidiosa*. The GM PDB was further genetically modified with a kill-switch, which is triggered in plants that are not infected by the pathogen. This kill-switch removes the GM bacteria from non-infected plants within several days.

#### 3.3.2. Exposure Pathways and ERA Considerations

Dissemination of the GM PDB and thus the GM bacteriophages will be achieved through the injection of the PDB into cultivated crop plants. Due to the GM chitin-binding protein, the GM bacteriophages will be dispersed by insect vectors, including species that spread *X. fastidiosa.* Relevant for the determination of exposure pathways are the insect vectors, which can transmit the PDP and/or the GM phages to other infected or non-infected host plants. Besides the primary vector, the highly polyphagous meadow spittlebug *Philaenus spumarius*, other insects feeding on infected host plants need to be considered as potential vectors of dissemination among trees and between orchards [88]. In the EU, climatic factors determine the establishment of the target pathogen. Suitable areas for a potential establishment differ between *Xylella* subspecies [88], but in general, horticultural plantations in southern Europe are the most relevant potential receiving environments. About 174 plant species have been identified as potential hosts for *X. fastidiosa*, including economically important plants like olive, grapevine, stone fruit, and forest trees [14,88].

The intended exposure pathways comprise the following routes:Artificial inoculation of horticultural trees (e.g., olive trees) infected with *X*. *fastidiosa* with the GM PDB and the spreading of the expressed GM phages within the plants’ xylem. Due to the kill-switch, the GM PDB is assumed to be present for just a few days in plants that are not infected by *X. fastidiosa*. In addition to the targeted pathogen, other bacteria residing in the xylem of the inoculated plants would be exposed, as would insects feeding on the infected plants;Spreading to other infected and non-infected horticultural plants by specific insect vectors, i.e., *Philaenus spumarius*, which is intended to spread the expressed GM phages, due to their ability to adhere to the chitin cuticula of the vector.In addition, several other (unintended) pathways of exposure need to be considered:Spreading by other insect vectors, e.g., cicadas, or other insects, such as xylem-feeding species, to non-target plants. The range of such insects that are able to spread the GM phages is considered to be broad due to the ubiquitous occurrence of chitin in insect cuticles; however, limited data are available to identify all the relevant insect vectors [88,89];The exposure of non-target bacteria in microbiomes associated with infected plants. Dependent on the ability of the GM phages to spread throughout the plants’ vascular system and possibly to the phyllosphere and the rhizosphere, non-target bacterial species from these microbiomes could be exposed.

For a risk assessment, the properties of the GM PDB as well as of the GM bacteriophages have to be considered. This poses additional challenges to identifying and characterizing the possible pathways to harm that may be triggered by the application. The following aspects need to be considered:The survival, spreading, and persistence of the PDB and the GM bacteriophage need separate consideration. The survivability and spreading of the GM phages, if released within the target plant, and their intended spreading by a vector insect are difficult to predict, as habitats other than those intended may be exposed to the GM phages. The persistence of the GM phages in plants is also not known. In general, phages are able to survive in planta for several weeks even in the absence of a bacterial host [86]. A theoretical model has been developed for predicting the spreading of the GM bacteriophages in the Italian region of Apulia based on data for *X. fastidiosa* [90,91]. The results showed that it might spread successfully across a large (10 × 10 km) area over a period of 40 days;Horizontal gene transfer (HGT) mediated by bacteriophages has been considered an environmental risk in other contexts (see González-Villalobos and Balcázar [92]). However, virulent (lytic) phages such as those used in the Xylencer application are expected to have a lower capacity for transduction of pathogenic factors [93]. The developers concluded that the transfer of the (inactive) Cas9 (dCas9) protein from the GM phages into the *X. fastidiosa* genome could have negative effects on the efficacy of the Xylencer approach. Due to the functions of the other transgenic elements, the risk for other unintended effects due to HGT is considered low;With regard to effects on target organisms, the assessment of the host specificity of the GM phages for the pathogen *X. fastidiosa* is considered an important aspect. The possible development of resistance to the GM phage treatment in the target organism also needs to be assessed. Any evolutionary changes in the virulence and pathogenicity of the GM phages may influence the effects on the target organism(s); however, they are challenging to assess. In addition, it is uncertain whether the predicted genetic changes, such as the loss of transgenic sequences, e.g., of the fusion gene between the phage’s capsid protein and the truncated chitin-binding protein, would indeed occur;Concerning effects on non-target organisms, the possible range of the host bacteria needs to be considered to determine whether non-target bacteria that occur in exposed plant microbiomes, including beneficial ones, may be affected. As shown by Ahern et al. [85], two of the employed phages are related to a larger group of virulent *Xanthomonas* phages, and the other two are related to phages occurring, e.g., in *Burkholderia* (phage AH2), Enterobacter (phage Enc34), other enteric bacteria (phage Chi), and *Providencia* (phage Redjac). The host range may also be changed by mutations, e.g., those affecting the receptor-binding proteins of the respective phages [94]. A possible reduction in beneficial bacteria may affect the overall performance and therefore the yield of the treated plants [95]. In addition, potential adverse effects of the transgenic proteins in exposed non-target insects or non-target plants need to be considered;The effects on the biogeochemical processes resulting from interactions of the GMO with target and non-target organisms if the bacteria of the soil microbiome are affected need to be considered. Some phages can persist for several weeks in the soil [96];Environmental impacts due to changes in the current management need to be considered, as the Xylencer application is meant to replace the current conventional, environmentally harmful management measures to control the target pathogen, i.e., the use of antimicrobial substances or pesticides to control the insect vector(s). Some existing measures may not be compatible with the release of GM phages or PDBs. Therefore, the use of a GM-phage-based approach must be closely coordinated and the efficacy monitored in order to avoid a failure to control the pathogen or subsequent harmful management measures.

Overall, considerable uncertainties exist regarding the long-term stability and interactions of the GM PDB and the GM bacteriophages in natural environments. The long-term efficacy of the approach is uncertain due to the dynamics of the interactions of phages with bacterial hosts, including the development of resistance. Uncertainties are also associated other aspects of the approach, including the loss of capacity for transmission of the GM phages by insects and the efficacy of the kill-switch engineered in the PDB. In this context, approaches have to be developed for the assessment of the long-term robustness and reliability of the novel genetic circuits. Thus, challenges exist for the prediction of evolutionary changes in the GM PDB and the GM phages for potential long-term and large-scale applications in long-lived organisms such as trees.

The confinement of the GM bacteriophages to their target environments is hard to ensure due to uncertainties regarding the spreading dynamics of the target pathogen and the GM bacteriophages as well as their host specificity. Substantial difficulties exist in defining the receiving environments and in assessing the consequences of the spreading of GM bacteriophages to other ecological niches and the resulting interactions in the environment. These uncertainties need to be addressed during the assessment and need to be considered regarding the surveillance and monitoring of the exposure and long-term effects.

## 4. Challenges for the ERA of GM Virus Applications in the EU

Several challenges for the risk assessment of GM virus applications in the EU are connected to the availability of specific guidance, specifically for the ERA of such applications, as confirmed by More et al. [78]. While the general principles provided in Dir. 2001/18/EC are deemed to be suitable, challenges remain regarding the appropriate approaches for the assessment of the risks of GM viruses, i.e., detailed guidance to address specific risk issues and the methods and endpoints that are applied during the assessment [78,96]. When updating relevant guidance documents, such as the guidance for the risk assessment of GMMs and their food and feed products [97], their scope has to be revised to better include GM virus applications.

Aspects covered by guidance from other regulatory fields, such as guidance published by the European Medicines Agency (EMA), need to be considered for this update. The EMA has published guidance addressing live recombinant-virus-vectored vaccines [98] and immunological veterinary medicinal products [99]. The available EMA guidance for the ERA of all veterinary medical products [100] is very general and outdated, e.g., the guidance related to the changes in animal husbandry systems since 1996, such as an increase in the average free-range pasture time [101]. An EMA concept paper for the revision of the guidelines on live recombinant vector vaccines for veterinary use [102] calls for a revision based on the requirements of Annex II of the Regulation (EU) 2019/6 for veterinary medical products, which again refers back to the requirements stipulated by Dir. 2001/18/EC. In the EU, the ERA for veterinary applications is conducted under the responsibility of member states authorities based on more detailed national guidance. Further exchange and harmonization of the national approaches thus seem necessary to guarantee uniform safety standards.

To improve the current situation, further specifications are required concerning a couple of different, albeit interconnected, aspects:General approaches for the assessment;The assessment of exposure of the environment;The assessment of molecular and phenotypic properties;Specific guidance for specific adverse effects (“specific areas of risk”) for the ERA according to Directive 2001/18/EC (Annex II, C.3).

### 4.1. Considerations Regarding General Approaches for the ERA

The current guidance for the ERA of GMOs recommends a comparative approach, as outlined, e.g., in the EFSA [98]. However, this approach may not be feasible for GM virus applications with a high level of novelty and complexity and in the absence of a comparator (e.g., a non-modified parental microorganism) that has already been evaluated for safety and has a known history of safe use. For applications such as GM CTV-SoD and the GM bacteriophages proposed for the Xylencer application, limitations in terms of using a comparative approach are apparent.

In this context, the QPS approach (qualified presumption of safety) may be an important tool [96]. Microbes that have a QPS status based on a pre-assessment, including their taxonomic identity, the body of available knowledge, and potential safety concerns, are presumed safe, not only for consumers but also for the environment [103]. However, for GM phages, the QPS approach is not considered applicable due to general difficulties regarding the taxonomic classification of bacteriophages, the distinction of transducing and non-transducing phages, and their ability to carry potentially harmful traits [96]. For GM viruses based on quarantine agents, e.g., the GM CTV-SoD application [97], no QPS status can be assigned either.

In these cases, a per se risk assessment needs to be carried out that does not rely on a comparison of the GM virus with its non-GM counterpart that is familiar, i.e., one that has been adequately characterized concerning its environmental safety [78]. Thus, extra data for the GM virus are required to assess its environmental effects; however, sufficient guidance for a per se assessment of GM virus applications is lacking.

### 4.2. Considerations Regarding Different Extents of Environmental Exposure

The range of GM virus applications is very broad and includes applications with a different spatial and temporal extent of environmental exposure. The different levels of exposure could, in turn, result in a range of environmental effects and a varying degree of uncertainty associated with a specific assessment.

At one end of the range are GM virus applications that are designed for intentional spreading and/or persistence in the environment (e.g., the Xylencer GM bacteriophage proposal or the transmissible vaccines that are currently being developed to reduce the Lassa fever virus reservoir in wild-living rodents using a contagious cytomegalovirus isolated from the host species as vaccine vector [40]). These applications have a complex or very complex design; limited information concerning the relevant risk issues is available for their assessment, and substantial uncertainties exist regarding their ability to spread in the environment and regarding the risks resulting from their release. Due to their transmissibility, they may cause transboundary movement incidents, and since they are considered to be non-retrievable, the possible options for their control are limited. Such applications pose many challenges for risk assessment and management and demand intense cooperation and coordination between different regulatory bodies and the applicants/developers. In some cases, such applications are developed in one country (e.g., the UK) but are meant to be released in another (e.g., African) country. This underscores the urgent need for international collaboration to exchange results and conclusions of risk assessments that address all the associated uncertainties.

For some of these applications, safe-by-design considerations were applied during their development. The safe-by-design concept was conceived to minimize the risks of emerging technologies, such as nanotechnology and biotechnology, e.g., by making safer design choices during the early stages of development [104]. The application of this concept has been discussed for several areas of biotechnology, including the development of GM crop plants [105] and vaccines [106]. For some of the GM virus applications reviewed in this study, safe-by-design considerations were implemented, e.g., the use of parental viruses with a narrow host range or mechanisms to induce the loss of transgenic constructs during propagation, as well as mechanisms, e.g., kill-switches, to limit the persistence of the GM organisms or the production of GM bacteriophages. However, it is uncertain whether such complex genetic mechanisms would function reliably after their environmental release under natural conditions and in different receiving environments. In case these mechanisms fail, unintended effects are more likely to occur.

The majority of GM virus applications for environmental release, however, represent applications with a limited or low ability to spread (including GM CTV-SoD or replication-deficient GM vector vaccines). As indicated in the discussion of the analyzed examples, challenges and uncertainties concerning their risk assessment also remain for these applications. This concerns issues such as genetic plasticity/stability, the persistence of the infectious material in the environment, the ability to regain replication-competence and transmissibility, changes in the host range or pathogenicity, and uncertainties regarding the capacity of the (insect) vector species’ transmissibility. The current information base (including information originating from other countries) for the assessment of all the relevant risk issues according to Dir. 2001/18/EC is insufficient and needs to be further developed [78]. Existing knowledge from risk assessments carried out for the contained use of GM viruses may be helpful, but it is certainly not sufficient for environmental applications.

Another group of applications of GM virus tools are only intended to be used under contained conditions (e.g., GM viruses for in vitro approaches to genetically modify plant or animal cells or to transiently express transgenes). Without underestimating the specific challenges in particular cases, these applications would generally present fewer difficulties for risk assessment.

### 4.3. Considerations Regarding the Assessment of Molecular and Phenotypic Properties

For the general molecular characterization of a GM virus, whole-genome sequencing (WGS), which is already mandatory for bacteria and yeast, should be used, as recommended by EFSA [78,96]. The aim of such WGS is to identify possible genes of concern, such as virulence or pathogenic genes, toxigenic factors, and to predict horizontal gene transfer. In addition, WGS can identify unintended genetic changes introduced by the GM methods used to develop the particular GM virus application [107]. WGS approaches have been recommended for the assessment of unintended genetic modifications in other GMOs, in particular, for genome-edited GM plants [108,109].

WGS is also important for the taxonomic identification of viruses, including bacteriophages. Currently, the guidance for the taxonomic identification of viruses, including bacteriophages is lacking; however, the considerations for the taxonomic identification of viruses in the QPS approach can be taken into account [110].

The stability of GM virus constructs, in particular, GM viruses based on RNA viruses, is a relevant issue for molecular characterization. WGS should be used to address sequence variability and its impact on biological activity [61,78]. In the framework of applications that employ more than one type of GM virus (such as in the case for the Xylencer application, which proposes using four different GM bacteriophages), all GM viruses (or bacteriophages) have to be assessed.

In addition to a full characterization of their genotype and stability, a phenotypic characterization of the particular GM viruses needs to be conducted, particularly regarding their toxigenicity and pathogenicity. The EFSA recommends that the existing microbiological EFSA guidance [111,112] be used regarding testing of the biological characteristics of GM viruses, such as their biological activity and host range [78]. However, suitable model systems for testing virulence and pathogenicity for non-target hosts in the context of the microbial characterization are lacking and need to be developed [78].

### 4.4. Considerations Addressing Specific Guidance for “Risk Areas” for ERA

In general, guidance as to how to assess the impact of GM viruses released into the environment regarding all the mentioned specific areas of risk in ERA is lacking and needs to be developed [78]. The examples of GM virus applications discussed in this study show that there is a need to define and specify endpoints for testing as well as appropriate testing methods for the ERA. Risk assessments conducted by other countries, e.g., in the USA for the GM CTV-SoD application, can be taken into account, but they need to be reviewed in terms of whether they address all the specific areas of risk according to Directive 2001/18/EC.

The key questions for the ERA of GM viruses relate to the assessment of their survival, spreading, and persistence in the receiving environments. Virus and bacteriophage populations are known to fluctuate significantly over time due to dynamic virus–host adaptation mechanisms [113]. For example, for bacteriophages, significant uncertainties exist, as several ecological models describing these interactions have resulted in different outcomes [114]. It is still unknown whether the use of a GM bacteriophage with novel traits would further complicate such assessments. The risk assessment needs to properly evaluate whether a GM virus is able to spread to and survive in novel, previously non-accessible habitats due to its characteristics and novel traits. In particular, for transmissible viruses or phages, it would be challenging to properly define host organisms, including target and non-target organisms.

For the assessment of the effects of GM viruses on non-target organisms, implications for protection goals, including ecosystem services, should be taken into consideration. Again, for applications with complex exposure patterns, defining and selecting relevant non-target organisms may be challenging. The EFSA has noted a particular lack of guidance for testing the virulence/pathogenicity of viruses [78].

The recent availability of metagenome data indicates that soil viruses are very abundant and highly diverse but largely uncharacterized [115]. Soil viruses, however, are highly relevant for important soil functions such as the biogeochemical cycling of carbon and nutrients in the soil environment, the microbial composition and activity of the soil, and the responses of the soil biota to environmental changes, e.g., higher soil temperatures or changes in soil moisture induced by climate change [116]. These dynamic interactions are only poorly understood and present challenges to assessing the direct and indirect impacts of GM bacteriophages on soil (micro)biota.

Another aspect that is related to challenges for both risk assessments and for considerations regarding the sustainability of GM virus applications is the efficacy of the individual applications, particularly for interventions that need to maintain a sufficient efficacy over longer periods. Examples of such interventions include GM virus applications designed to protect or restore the health of long-lived plants, e.g., orange or olive trees. Similar considerations were taken for bacteriophage-based treatments of foods of animal origin [117]. This requires an assessment of the occurrence of resistant pathogenic bacteria and their pathogenicity (e.g., virulence). For environmental applications, the development of resistance in pathogenic target organisms may compromise important protection goals, including the loss of ecosystem services, e.g., with regard to an altered susceptibility of the target plants to pathogens or the creation of novel plant diseases (see Directive 2001/18/EC).

## 5. Conclusions

The horizon scan described in this study and the analysis of several examples of GM virus applications indicate that, regarding the risk assessment and risk management of GM virus applications, a number of significant challenges exist at present.

To increase the necessary preparedness in the EU and at a global level, further horizon-scanning activities to identify emerging GM virus applications are necessary. Previous activities in this field have not provided a comprehensive overview of the ongoing and future developments, e.g., those concerning transmissible GM virus vaccines for which in-depth information is not readily available. In addition to the analysis of the relevant scientific literature, expert-driven processes at an international level for information gathering and exchange are considered important. These activities should include experts who develop GM virus applications as well as regulatory and risk-assessment experts.

Additionally, the existing guidance for risk assessment, particularly the guidance for the ERA of GMMs, needs to be updated to include GM viruses. The overview of the potential GM virus applications presented in this study indicates that applications are developed for different purposes and subject to different regulations. Thus, it is important that the guidance that is available in different regulatory areas is reviewed and consolidated for easier accessibility and coordination of risk assessment requirements. This requires the involvement and cooperation of experts from the respective fields.

A further important aspect regarding the oversight of GM viruses is that for the post-market environmental monitoring of GM virus applications in the EU, no established concepts currently exist. Due to their high potential for survival and spreading as well as their ability to quickly mutate and evolve, adequate monitoring is very important for GM virus applications. Furthermore, robust monitoring approaches are important to provide data for the further assessment (or re-assessment) and management of GM virus applications and should be developed in the near future based on experience with recently established virus-surveillance approaches.

Regarding the management of GM virus applications, it seems necessary to identify inherent riskier applications, e.g., those involving GM viruses that are able to spread intentionally or unintentionally, as compared to applications with less risky characteristics. Available interventions with a lower risk should be prioritized in case such alternatives exist.

## Figures and Tables

**Figure 1 ijms-25-01507-f001:**
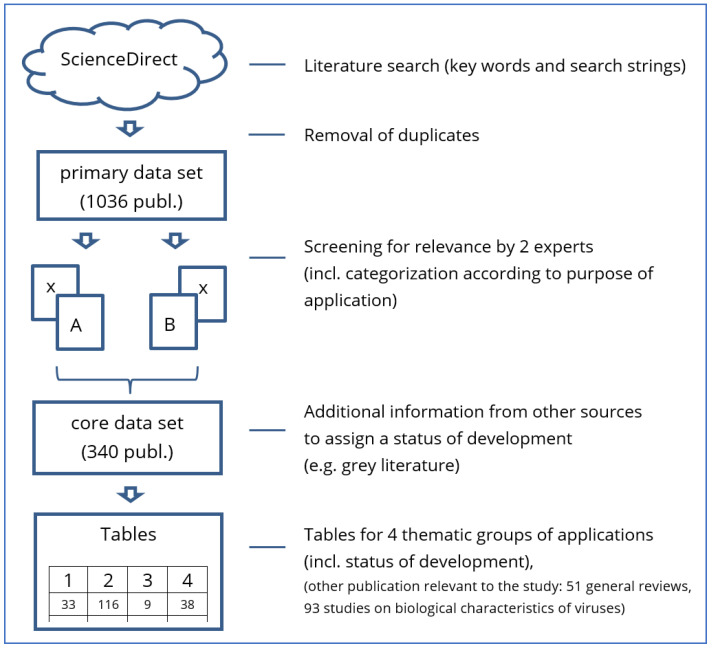
Workflow for the literature survey conducted during this study (publ.: publications).

**Figure 2 ijms-25-01507-f002:**
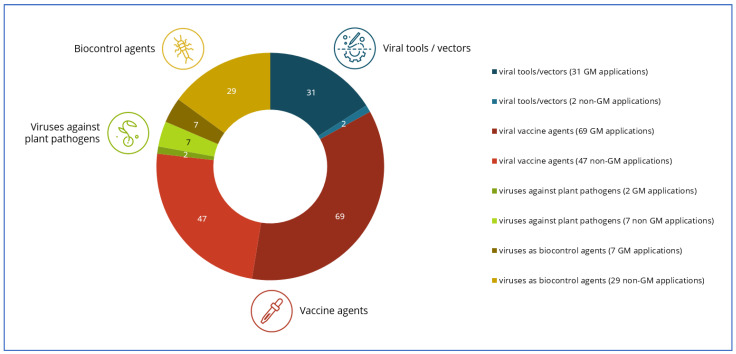
Overview on the four major areas of viral applications identified in the literature survey. Publications on viral tools (vectors) are indicated in blue color (dark blue: GM applications, light blue: non-GM applications); publications dealing with vaccine agents are indicated in red color (dark red: GM applications, light red: non-GM applications); publications on viruses against plant pathogens in green color (dark green: GM applications, light green: non-GM applications); and publications on viruses as biocontrol agents are indicated in brown color (dark brown: GM applications, light brown: non-GM applications).

**Table 1 ijms-25-01507-t001:** Numbers of publications identified as relevant for the different groups of applications and of publications featuring GM virus applications. The last three columns indicate the state of development of these GM virus applications.

Thematic Group	Relevant publ. ^1^	GM appl. ^1^	BR ^1^	R&D ^1^	NM/M ^1^
GM viral tools (vectors)	33	31	3	23	5
GM vaccine agents	116	69	15	51	3
GM viruses against plant pathogens	9	2	-	2	-
GM viruses as biocontrol agents	36	7	6	1	-

^1^ publ.: publications; appl.: applications; BR: basic research; R&D: research and development stages; NM/M: near-market/market applications.

## Data Availability

The data (literature references) discussed in this study are available in the Appendix A to this article.

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
