# Peer review of "Scanning the Horizon for Environmental Applications of Genetically Modified Viruses Reveals Challenges for Their Environmental Risk Assessment"

_ijms, 2024, doi:10.3390/ijms25031507_

Round 1

Reviewer 1 Report

Comments and Suggestions for Authors

This comprehensive review/perspective/scan describes current GM virus applications in agriculture, veterinary, and nature conservation and then describes the challenges for risk assessors and regulators. The authors emphasize that ongoing efforts are required to monitor and identify emerging GM virus applications. It is crucial to develop effective approaches for risk assessment and management, specifically addressing challenges related to their environmental release, potential transmission, and adverse effects. However, the current readiness at the EU and international levels to evaluate such applications is limited. This study tackles these challenges by conducting a horizon scan to identify relevant GM virus applications and addressing outstanding issues in environmental risk assessment (ERA) through an evaluation of case study examples, highlighting the lack of scientific information and appropriate guidance for ERA. 

When describing the first example scenario, it may be good to mention that recombination of viral vectors containing immunogenic peptides will likely not be retained within the environment if it has a negative impact on viral replication (i.e. eliciting an immune response).  This is actually a strength of the approach and will limit environmental spread of the vector/transgene.  Unless the insert is beneficial to the virus, it will likely get removed during sequential viral replication/hybridization events if they occur. 

The article is well written, pertinent to today’s current GM research environment and to policy makers.  A few grammatical items should be checked below; however, no major concerns with the current draft.

·       For clarity to general audience, please write out Genetically Modified in the title and define ‘GM’ at the first instance within the abstract as well as the introduction.

·       Page 3, Line 99 “The study at hands” should be “The study at hand”

·       Page 3, Line 111 “gray literature” should be “grey literature”;  repeated in Page 5, Lines 183 & 185, Page 7 Line 273

·       Page 4, Line 142 “transformation vectors” should be “transduction vectors”

·       Page 5, Line 204 remove extra comma “applications.,”

·       Page 6, Line 206 add space between words “animalsge-

·       Page 7, Line 264 add comma “for dogs cats, sheep and rabbits”

·       Check line/space formatting on Page 9 under section 3.1.1.   Some paragraphs begin with a tab and justified while other paragraphs are not.

·       Page 10, Line 448 “or even month” should be “or even months

·       Page 17, Line 774  remove extra period “control are limited..”

Comments on the Quality of English Language

Please see above for items to be checked.

Author Response

Dear Reviewer

Kind thanks for your thorough review of our manuscript and for your overall supporting evaluation. All of your suggestions and corrections have been included in the revised manuscript.

The abbreviation GM was spelled out in the title and introduced in the abstract.

The description of the first example scenario was amended according to your suggestion to indicate that the environmental spread of viral vectors expressing immunogenic peptides will be limited and transgenes which are not associated with a selective advantage will likely be lost. The factors which influence such genetic instability are indicated as well as the challenges to predict such evolutionary changes (see lines 471-481).

All editorial or grammatical errors which were indicated in your review have been corrected accordingly.

In addition to the revisions recommended in your review a few additional editorial errors (missing blanks, typos) were corrected. All footnotes referring to internet resources have been changed into references to the respective internet documents and included in the list of references.

According to suggestions of the second reviewer a new figure indicating the major areas of application of GM viruses was included (Line 150). In addition the challenges for the assessment of changes to the soil virome were mentioned (lines 885-892).

Reviewer 2 Report

Comments and Suggestions for Authors

This is an excellent review focusing on potential environmental applications of GM viruses and related risk based on literature survey. The review is very well described, and comprehensive. The sections are logically organized along with detailed methodologies followed in the review process. Both advantages and limitations associated with the use of GM virus specifically in agriculture are well addressed.

Comments

Line 119. Please answer, why only Science Direct was use for the literature search, why not PubMed and/or Web of Science?? Using other database as well might have given more robust findings. With wider coverage..

Line 135, what were the qualifications of those two individual experts?

It will be good if major areas of application of GM viruses are presented in a figure/schematically/with a diagram pic for easy quick understanding my the readers.

Comments on the Quality of English Language

Minor editing of English language required

Author Response

Dear Reviewer

Many kind thanks for your review and supportive feedback. We are happy to answer the questions raised in your comments:

Q1: (Line 119) Science Direct was used for practical reasons connected to licencing and accessability issue to avoid rights infringements when using search tools or accessing literature. This is an important issue nowadays and unfortunately limits easy access to literature as well as search tools. We are aware of the limitations and thus refrained from using terms like “systematic review” to describe our approach. To further emphasize this we include the expression “non-exhaustive survey of scientific literature” in Line 110. We are however convinced that the main findings regarding the major areas of application and their stage of development are nevertheless sound. Since some relevant information on applications is not documented in scientific literature, but rather in grey literature, regulatory information and company information we do not believe that a more comprehensive literature survey would have changed our results and conclusions in a significant way. We also indicate that further horizon scanning involving developers and regulatory experts at the international level is necessary to provide a comprehensive overview of ongoing and future developments (see L 908ff).

Q2: (Line 135) Both experts have an academic background in biology, one of them with a masters degree in ecology the other with a PhD in molecular genetics and microbiology and 2yrs experience at the Institute of Virology of the University of Vienna (now: Vienna Medical University). Both experts also have ample experience (20 yrs +) in environmental risk assessment of GMOs and are involved in ongoing research concerning emerging GM applications in plants, animals and microorganisms. We included a reference to the background of the experts in line 135 for explanation.

Suggestion 3: We included a graphical illustration indicating the major areas of application of GM viruses to complement the text and tables (Line 150). Kind thanks for your suggestion!

In addition to the revisions based on your review a number of editorial and grammatical errors were corrected. All footnotes referring to internet resources have been changed into references to the respective internet documents and included in the list of references.

According to suggestions of the other reviewer an addition was included to the description of the first example scenario to indicate that the environmental spread of viral vectors expressing immunogenic peptides will be limited and transgenes which are not associated with a selective advantage will likely be lost. (see lines 471-481). In addition the challenges concerning the assessment of changes to the soil virome were mentioned in lines 885-892.